# Utilization of a Wheat50K SNP Microarray-Derived High-Density Genetic Map for QTL Mapping of Plant Height and Grain Traits in Wheat

**DOI:** 10.3390/plants10061167

**Published:** 2021-06-08

**Authors:** Dongyun Lv, Chuanliang Zhang, Rui Yv, Jianxin Yao, Jianhui Wu, Xiaopeng Song, Juntao Jian, Pengbo Song, Zeyuan Zhang, Dejun Han, Daojie Sun

**Affiliations:** 1College of Agronomy, Northwest A&F University, Xianyang 712100, China; lvdongyun1102@163.com (D.L.); wyanan@nwafu.edu.cn (C.Z.); 18404969211@163.com (R.Y.); bilwqz@163.com (J.Y.); wujh@nwafu.edu.cn (J.W.); wheat5200stem@163.com (P.S.); 18238768351@163.com (Z.Z.); 2Zhumadian Academy of Agricultural Sciences, Zhumadian 463000, China; weiduxp@163.com; 3Nanyang Academy of Agricultural Sciences, Nanyang 473000, China; jjt312024501@163.com

**Keywords:** wheat, plant height, grain traits, Wheat50K, genetic map, QTL

## Abstract

Plant height is significantly correlated with grain traits, which is a component of wheat yield. The purpose of this study is to investigate the main quantitative trait loci (QTLs) that control plant height and grain-related traits in multiple environments. In this study, we constructed a high-density genetic linkage map using the Wheat50K SNP Array to map QTLs for these traits in 198 recombinant inbred lines (RILs). The two ends of the chromosome were identified as recombination-rich areas in all chromosomes except chromosome 1B. Both the genetic map and the physical map showed a significant correlation, with a correlation coefficient between 0.63 and 0.99. However, there was almost no recombination between 1RS and 1BS. In terms of plant height, 1RS contributed to the reduction of plant height by 3.43 cm. In terms of grain length, 1RS contributed to the elongation of grain by 0.11 mm. A total of 43 QTLs were identified, including eight QTLs for plant height (PH), 11 QTLs for thousand grain weight (TGW), 15 QTLs for grain length (GL), and nine QTLs for grain width (GW), which explained 1.36–33.08% of the phenotypic variation. Seven were environment-stable QTLs, including two loci (*Qph.nwafu-4B* and *Qph.nwafu-4D*) that determined plant height. The explanation rates of phenotypic variation were 7.39–12.26% and 20.11–27.08%, respectively. One QTL, *Qtgw.nwafu-4B*, which influenced TGW, showed an explanation rate of 3.43–6.85% for phenotypic variation. Two co-segregating KASP markers were developed, and the physical locations corresponding to KASP_AX-109316968 and KASP_AX-109519968 were 25.888344 MB and 25.847691 MB, respectively. *Qph.nwafu-4B*, controlling plant height, and *Qtgw.nwafu-4B*, controlling TGW, had an obvious linkage relationship, with a distance of 7–8 cM. Breeding is based on molecular markers that control plant height and thousand-grain weight by selecting strains with low plant height and large grain weight. Another QTL, *Qgw.nwafu-4D*, which determined grain width, had an explanation rate of 3.43–6.85%. Three loci that affected grain length were *Qgl.nwafu-5A*, *Qgl.nwafu-5D.2*, and *Qgl.nwafu-6B*, illustrating the explanation rates of phenotypic variation as 6.72–9.59%, 5.62–7.75%, and 6.68–10.73%, respectively. Two QTL clusters were identified on chromosomes 4B and 4D.

## 1. Introduction

Wheat (*Triticum aestivum* L.) is a major food crop globally, providing carbohydrates and protein for 35% of the global population. It is estimated that wheat production will increase by more than 70% in the next 30 years to meet the needs of the growing population [1]. To ensure global food security, genetic improvement of food production will be one of the main goals of wheat breeding programs [2,3,4].

Both 1000-grain weight (TGW) and the genetic improvement of related traits, which play a vital role in wheat yield, are applicable to increasing wheat yield. TGW is mainly affected by grain morphological parameters, such as grain length and grain width [4,5,6]. TGW-related genes, including sucrose synthase genes, encode cell wall invertase and cytokinin oxidase/dehydrogenase. The sucrose synthase genes *TaSus1-7A*, *-7B* and *TaSus2-2A*, *-2B* determine TGW and grain size [7,8], *TaGW2-6A*, *-6B* the grain width [9,10], and *TaGS-D1* the grain size [11]. *TaCwi-A1* encodes cell wall invertase [12], *TaCKX6-D1* encodes cytokinin oxidase/dehydrogenase [13], and *TaGASR-A1* is a putative Snakin/GASA protein associated with grain length (GL) (Dong et al., 2014). The inheritance of grain traits is relatively stable, forming a higher heritability than overall yield [14]. The method is suitable for QTL analysis of wheat samples planted and collected from different places and years, and a stable QTL can be retrieved and detected. Over the past 20 years, more than 150 QTLs related to TGW, grain length, and grain width have been identified, which are distributed on 21 chromosomes of wheat [5,15,16,17,18,19,20,21,22,23,24,25,26,27,28,29,30,31,32,33,34,35,36,37,38,39,40,41,42,43,44,45]. Some studies have shown that there is a significant positive correlation between plant height and TGW [19,32,33,39,46,47]. The application of *Rht1* (*RHT-B1b*) and *Rht2* (*RHT-D1b*) in the 1960s set off a green revolution in wheat breeding. So far, 25 *Rht* genes have been identified in wheat [48,49]. Amongst these 25 genes, *Rht1* and *Rht2* are dwarfing genes that show insensitivity to gibberellins located on chromosomes 4BS and 4DS, respectively [13]. The wild alleles *Rht-B1a* and *Rht-D1a* also have a significant positive correlation with TGW [32,50]. Another gene, called *Rht8*, is sensitive to gibberellins for reducing plant height and is located on the 2DS chromosome. *Rht8* is another widely applied dwarfing gene that has no obvious negative effect on TGW, but affects panicle length. Thus, *Rht8* is a typical pleiotropism gene [6,51]. The genetic relationship can be investigated by targeting gene loci related to TGW and plant height, obtained by QTL mapping [30,52].

QTL genetic mapping is a crucial means to analyze functional loci [28]. Constructing a saturated genetic map is the key to QTL mapping, and molecular markers are the genetic map carrier. *Triticum aestivum* L. is a typical allohexaploid (AABBDD) composed of three subunits, and it represents the largest crop genome. Moreover, it is also the genome with the highest proportion of repetitive sequences such as transposable elements (84.7%) (IWGSC2018). Multitudes of SNP markers bear abundant polymorphism [53], and mapping results are quite advantageous in terms of accuracy and precision, especially for QTL mapping of quantitative traits [53,54]. By constructing a high-density genetic map to target the SNPs’ genetic and physical loction, collinearity analysis is performed, and then the recombination rate in different regions of the chromosome can be judged. After comparing the genetic and physical distances between adjacent markers, the relative changes of recombination rates in each chromosome can be further investigated and analyzed. The range of the mating population required for a recombination event in a specific region can be estimated. Scientific and accurate estimation for breaking the chain of specific target areas can be provided, and accurate judgments for evaluating genetic linkage drag, together with guidance for improving breeding efficiency, can be achieved [54,55].

Until now, couples of common wheat SNP microarrays, including Wheat9K [56], Wheat90K [37,57,58], Wheat820K [59], Wheat660K (http://bioservices.capitalbio.com/index.shtml) [37,57,58], and the Wheat55K SNP array, have been developed based on the 660K SNP array [60,61,62,63,64,65].The Wheat50K SNP array is a high-efficiency genotyping technology completed by the Institute of Crop Science of the Chinese Academy of Agricultural Sciences and Affymetrix. The technology was developed using high-quality SNP markers selected from Wheat90K SNP arrays, 660K SNP arrays, and 35K SNP arrays. In the 50K SNP array, there are 135 functional markers and 700 SNP markers closely linked to known QTLs [66]. The functional markers covering ten TGW-related genes and two plant height-related genes are shown in Appendix A.

In this study, a Kompetitive allele-specific PCR (KASP) marker was used, which is a polymerase chain reaction-based (PCR) technology using fluorescence for single nucleotide polymorphism (SNP) and small insertion and deletion (InDel). KASP markers have the advantage of a low error rate and a relatively low cost compared to other SNP genotyping platforms such as TaqMan systems. According to the method of Ma et al. [63], SNPs located in the main QTL interval were selected to develop KASP markers.

This project aims to determine the chromosome recombination rates in different regions using collinearity analysis of the genetic positions and physical locations of the SNP markers. By mapping the environment-stable QTL region of grain-related traits, whether corresponding loci are located in the recombination-rich or recombination-barren area can be confirmed, and a reasonable judgment for further fine mapping can be fulfilled. By traits and linkage analysis of the relationship between plant height and grain traits, useful insights for the next steps of molecular breeding can also be provided.

## 2. Results

### 2.1. Agronomic Traits Analysis

As was shown in Table 1, significant differences when *p* = 0.01 in the four environments appeared in relation to the plant height, TGW, grain length, and grain width of the two-parent materials. In Table 1 and Appendix A, we can see that fluctuations occurred in the same traits in different environments, indicating that these four traits were easily affected by the environment. The agronomic traits failed to accord with a strictly normal distribution (*p* < 0.05). The trait heritability values of plant height, TGW, grain length, and grain width were 0.73, 0.62, 0.61, and 0.72, respectively. As can be seen, those of plant height and grain width were relatively high.

As was shown in Appendix A, there was a significant positive correlation between the same traits and different environments when *p* = 0.001. The correlation between different years in the same place was higher than that in other combinations, indicating that a high degree of environmental similarity was present in the same place but in different years. The correlation between plant height and grain length was negative, but there was a significant positive correlation between TGW and grain width. TGW had a significant positive correlation with the other three traits, and a higher correlation with grain width than that with other traits. The correlation between grain length and grain width was different in different environments.

### 2.2. Construction of a Genetic Map

#### 2.2.1. Description and Illustration of a Genetic Map

66,832 markers were subject to polymorphism analysis of population genotype by 50K gene microarray. A total of 19,601 SNP markers with differences were screened in the derived RIL populations of Xinong1376 and Xiaoyan81, while the remaining 15,822 markers were filtered by Chi-square test, and redundant markers were eliminated using the bin function of IciMapping. A total of 3136 bin markers, including 15,576 SNP markers, were eventually anchored to the genetic map. In addition, the genotyping, polymorphism marker, data filtering, physical map, genetic map, and bin map are all shown and illustrated in Appendix A. Based on the 660K chip labeling, the SNP markers that differed between the two parents were detected and stored in Appendix A. The total length of the linkage map was 4512.79 cM, the average map distance was 1.44 cM, and the maximum gap was 26.86 cM, which covered 21 wheat chromosomes. According to linkage lengths in the homologous groups, their sequence in descending order was the fifth, the seventh, the third, the second, the fourth, the sixth, and the first. The linkage lengths were 813.14 cM, 794.35 cM, 703.96 cM, 631.98 cM, 563.99 cM, 537.27 cM, and 468.12 cM, and the numbers of bin markers were 621 (2947 SNP markers included), 549 (2193 SNP markers included), 524 (2846 SNP markers included), 327 (1865 SNP markers included), 393 (2002 SNP markers included), 372 (1865 SNP markers included), and 272 (2016 SNP markers included), respectively.

The numbers of bin markers located in wheat A, B, and D chromosome groups were 1231, 1197, and 708, respectively. The linkage lengths were 1703.69 cM, 1298.23 cM and 1510.87 cM, and the average map distances were 1.38 cM, 1.08 cM, and 2.13 cM, respectively. Molecular markers in the D genome were no more than those in the other two subgroups. In addition, the longest linkage group corresponding to chromosome 3A was 312.11 cM, and the shortest corresponding to chromosome 1D was 130.85 cM. The maps of each linkage group were shown and illustrated in Table 2 and Appendix A.

#### 2.2.2. Collinearity Analysis of the Genetic Map

In this research, the genetic map and the collinearity map of the reference genome were analyzed as follows: The whole chromosome was included in the genetic map, the genetic map and the physical map were collinear, and the linkage map and the physical map were not linearly related. The recombination exchange on chromosomes was unbalanced, and the collinear diagrams of other chromosomes except for chromosome 1B appeared by and large S-shaped. The genetic positions of chromosomes increased linearly with the increase in physical locations, and the rest of the genetic positions aligned constantly with the increase in physical locations. This indicated that the two ends of the chromosome were recombination-rich areas and that the middle region was a recombination-barren area. A significant correlation of the genetic map and the physical one appeared when *p* = 0.001, the correlation coefficient ranged from 0.63 to 0.99, and the correlation coefficient of chromosome 1B was 0.63. The distribution presentation of bin markers on the reference genome showed that the number of bin markers on both ends of the chromosome was significantly higher than that of the middle region. The recombination rate of the two sides with a U-shaped distribution was significantly higher than that of the middle region, which confirmed that the ends of the chromosome were recombination-rich areas and the middle was the recombination-barren area. The reason for these findings was the inhibitory effect of centromere recombination.

No markers could be detected in the middle regions (more than 200 MB) of chromosomes 1D, 5A, and 6A. However, the linkage group was not divided into two parts in these regions, which were supposed to be recombination-barren regions. For nine chromosomes (2D, 3D, 5A, 5B, 5D, 6A, 6D, 7A and 7D), each chromosome included two linkage groups. For different linkage groups corresponding to the same chromosome, the grouping regions all appeared at both ends of the chromosome as the recombination-rich area, and the physical distance between the markers was less than 30 MB.

The collinearity map of chromosome 1B from 0 to 480 MB presented as an L-type curve. Although the gradual numerical values of physical location increased, the genetic distances were almost unchanged, and thus homologous recombination hardly occurred in the region. Xinong1376 belonged to 1BL/1RS translocation line, 1RS and 1BS hardly recombined, and the centromere’s inhibition of recombination happened in the middle region, making the collinearity map L-shaped.

#### 2.2.3. Effects of 1B/1R on Traits Related to Plant Height and TGW

1RS specific marker was used to detect the population, the strains containing 1RS and 1BS were 51 and 147, respectively, and the *p* value of the chi-square test was 8.95 × 10^−12^, which proved to be a severely segregated marker that couldn’t be linked to the linkage group. According to the typing of the specific markers, the unpaired data T test was performed on the traits related to plant height and TGW, and there was no significant difference between 1RS and 1BS. According to the typing of specific markers, a two-factor analysis of variance was performed on the agronomic traits, and the TGW and grain width were not affected by the genotype. According to the results of the variance analysis, Duncan’s new multiple range test comparison of plant height and grain length was conducted. In terms of plant height, 1RS contributed to the reduction of plant height by 3.43cm. In terms of grain length, 1RS contributed to the elongation of grain by 0.11mm (shown in Appendix A).

### 2.3. QTL Mapping Analysis

A total of 43 QTLs for PH, TGW, GL, and GW were identified by QTL mapping analysis (Table 3 and Appendix A). These QTLs with LOD values ranging from 2.51 to 53.34 were distributed on 15 chromosomes and explained 1.36–33.08% of the phenotypic variation (Table 3 and Appendix A). There were 8, 11, 15, and 9 QTLs detected for PH, TGW, GL, and GW, respectively (Table 3 and Appendix A).

Inclusive composite interval mapping (ICIM) for PH identified a total of eight QTLs, which were located on six different chromosomes (Table 3 and Appendix A): 2D(2), 4B, 4D, 5B, 5D, and 6B(2). The QTL on 4B, *Qph.nwafu-4B*, was detected in four environments. *Qph.nwafu-4B* was thus treated as a major QTL, which explained 9.32–13.76% of phenotypic variance with LOD values ranging from 7.93 to 26.85. As was expected, the positive allele of *Qph.nwafu-4B* was contributed by Xiaoyan81 (Table 3 and Appendix A). The QTL on 4D, *Qph.nwafu-4D*, was detected in each of four environments. *Qph.nwafu-4D* was thus treated as a major QTL, which explained 20.11–27.09% of phenotypic variance with LOD values ranging from 16.78 to 42.21. As we expected, the positive allele of *Qph.nwafu-4D* was contributed by Xinong1376 (Table 3 and Appendix A).

One QTL, *Qph.nwafu-2D.1*, for PH was detected in two environments, which explained 3.3–3.73% of phenotypic variance. The remaining QTLs were detected only in a single environment (Table 3 and Appendix A).

ICIM for TGW identified a total of eleven QTLs, which were located on eight different chromosomes (shown in Table 3 and Appendix A): 2A, 2B, 3A, 4B, 4D(2), 5A, 5D(3), and 6A. The QTL on 4B, *Qtgw.nwafu-4B*, was detected in three environments. *Qtgw.nwafu-4B* was thus treated as a stable QTL, which explained 3.43–6.85% of phenotypic variance with LOD values ranging from 2.85 to 4.37. As was expected, the positive allele of *Qtgw.nwafu-4B* was contributed by Xinong1376 (shown in Table 3 and Appendix A). Based on the initial QTL mapping results, we developed two KASP markers, KASP_AX-109316968 and KASP_AX-109333198, and integrated them into the genetic map. When remapping with this integrated KASP marker, it was indicated that *Qtgw.nwafu-4B* was located in a 5 cM interval on chromosome arm 4BS, between the markers of AX-111494900 and AX-94438527, containing the newly developed KASP markers, including KASP_AX-109316968 and KASP_AX-109333198 (Appendix A).Three QTLs, *Qtgw.nwafu-4D.1*, *Qtgw.nwafu-5A*, and *Qtgw.nwafu-5D.1*, for TGW were detected in each of two environments, which explained 2.85–14.79% of phenotypic variance. The remaining QTLs were detected only in a single environment (Table 3).

ICIM for GL identified a total of fifteen QTLs, which were located on ten different chromosomes (Table 3 and Appendix A): 1A, 1B(2), 3A, 4A, 4B(2), 4D, 5A, 5B, 5D(4), and 6B. The QTL on 6B, *Qgl.nwafu-6B*, was detected in four environments. *Qgl.nwafu-6B* was thus treated as a major QTL, which explained 6.68–10.73% of phenotypic variance with LOD values ranging from 3.35 to 8.79. As was expected, the positive allele of *Qgl.nwafu-6B* was contributed by Xinong1376 (Table 3). The QTL on 5A, *Qgl.nwafu-5A*, was detected in in three environments. Qgl.nwafu-5A was thus treated as a stable QTL, which explained 6.72–9.59% of phenotypic variance with LOD values ranging from 5.8 to 6.93. As we expected, the positive allele of *Qgl.nwafu-5A* was contributed by Xinong1376 (Table 3 and Appendix A). The QTL on 5D, *Qgl.nwafu-5D.2*, was detected in three environments. *Qgl.nwafu-5D.2* was thus treated as a stable QTL, which explained 5.62–7.75% of phenotypic variance with LOD values ranging from 4.36 to 6.13. As was expected, the positive allele of *Qgl.nwafu-5D.2* was contributed by Xinong1376 (Table 3 and Appendix A). Four QTLs, *Qgl.nwafu-1B.2, Qgl.nwafu-3A*, *Qgl.nwafu-4A*, and *Qgl.nwafu-4B.2*, for GL were detected in two environments, explaining 3.51–6.13% of phenotypic variance. The remaining QTLs were detected only in a single environment (Table 3 and Appendix A).

ICIM for GW identified a total of nine QTLs, which were located on seven different chromosomes (Table 3, Appendix A): 2B, 2D, 3A, 4B(3), 4D, 5D, and 6D. The QTL on 4D, *Qgw.nwafu-4D*, was detected in each of the four environments. *Qgw.nwafu-4D* was thus treated as a major QTL, which explained 6.32–12.12% of phenotypic variance with LOD values ranging from 3.2 to 7.93. As we expected, the positive allele of *Qgw.nwafu-4D* was contributed by Xinong1376 (Table 3). One QTL, *Qgw.nwafu-4B.1*, for GW was detected in two environments, which explained 6.85–6.95% of phenotypic variance. The remaining QTLs were detected only in a single environment (shown in Table 3 and Appendix A).

Two QTL clusters were identified on chromosomes 4B and 4D (Table 3 and Appendix A). For the QTL cluster on chromosome 4B, *Qtgw.nwafu-4B* for TGW was co-localized with *Qgl.nwafu-4B.1* for GL, and *Qph.nwafu-4B* and *Qgl.nwafu-4B.2* for GL were co-localized with *Qgl.nwafu-4B.2* and *Qgl.nwafu-4B.3* for GL in a region ranging from 51 cM to 77 cM. On chromosome 4D, *Qph.nwafu-4D* for PH was clustered with *Qtgw.nwafu-4D.1* for TGW, and *Qgw.nwafu-4D* for GW was clustered with with the alleles from Xiaoyan81 increasing PH, TGW and GW.

## 3. Discussion

### 3.1. The Impact of Linkage Map on QTL Mapping

In this research, a linkage map, based on 50K microarray markers, was constructed from 198 F_8_ RIL lines derived from the combination of two parents, Xinong1376 and Xiaoyan81. The linkage map had a total length of 4512.79 cM, covering 21 chromosomes of wheat. The reason why no marks could be targeted in the regions of more than 200 MB in the middle of the four chromosomes 1D, 5A, and 6A was that a recombination-barren area near the centromere appeared in the above regions, as was shown in Figure 1. Both parents were derived from the backbone parent Xiaoyan6, and a region with the same haplotype was formed rapidly [68], so that the two parents had no markers with polymorphic differences in the above regions. There was a long, excellent haplotype segment on chromosome 6A [60,69].

In this study, 43 QTLs were located. The genetic distance confidence interval was 0.5–12.5 cM, and the physical distance of the markers on both sides was 0.0201 MB–414.88328 MB. As was shown in Table 2, the genetic distance confidence interval was not proportional to the physical distance, which reflected the imbalance of the recombination exchange on the chromosomes.

By combining Appendix A and Figure 1, it appeared that there were 5 QTLs located in the recombination-barren region of the reference genome, and more than 20 MB QTLs were distributed in this candidate region. The linkage interval of *Qgl.nwafu-1B.1* was 0–0.5 cM, while the physical interval was 59.47117 MB–94.978091 MB and the interval physical distance was 35.506914 MB. The reason was that Xinong1376 belonged to the 1BL/1RS translocation line, and there was almost no recombination or recombination disorder between 1RS and 1BS [6,38,70,71]. Although the genetic distance of the confidence interval was short, the corresponding physical distance of it was far. As was shown in Appendix A, the linkage region of *Qtgw.nwafu-2B* was 95.5 cM–106.5 cM, and no marks could be targeted in this region. This area belongs to the reorganization cold spot area, and the corresponding physical distance was 153.585606 MB–568.468886 MB. The linkage regions of *Qtgw/gl.nwafu-3A*, *Qgl.nwafu-4A*, and *Qgw.nwafu-4B.2* were 132.5 cM–134.5 cM, 46.5 cM–53.5 cM, and 67.5 cM–69.5 cM, respectively, and the corresponding regions were 457.796943 MB–431.074614 MB, 407.389107 MB–129.089816 MB, and 114.952789 MB–161.548436 MB, respectively. As was shown in Table 2, the above three QTLs all fell in the recombination-barren region of linkage groups with a large physical interval. The confidence interval of *Qgw.nwafu-6D*, which was the largest, was 0 cM–12.5 cM, but the corresponding physical region was 12.650045 MB–8.255713 MB, and the interval was only 4.4 MB. *Qgw.nwafu-6D* was located at the top of the chromosome, and belonged to the recombination-rich region, with a big genetic distance but a short corresponding physical distance.

### 3.2. Comparison with Previous Research Results

Two loci as environment-stable QTLs, targeted in three or four kinds of environments, were *Qph.nwafu-4B* and *Qph.nwafu-4D*, which control plant height. In the confidence interval, the function markers including *Rht-1* and *Rht-2* were AX-179477460 and AX-86170701, respectively. According to the additive effect, the effect of the *Qph.nwafu-4D* mutant in lowering plant height was stronger than that of the *Qph.nwafu-4B* mutant, which was consistent with the results of Zhai et al. [6] The locus, *Qgl.nwafu-5A*, which controlled the grain length, corresponded to the physical location of 698.508129 MB–700.34701 MB, which was located at the end of the chromosome. Compared with the results of previous studies [23,29,30,31,32,33,34,35,36,37,38,39,40,41,42], Qgl.nwafu-5A was a new QTL. The location of *Qgl.nwafu-5D.2* which controlled the length of the grain corresponded to the physical location of 370.135626 MB-386.126855 MB. Based on previous research [22,24,35,42,43], *Qgl.nwafu-5D.2* was defined as a new QTL as well. The location of *Qgl.nwafu-6B*, which controls grain length, corresponded to the physical location of 704.884934 MB–718.376276 MB. Compared with the results of previous studies [35], the physical location marked by IWB2746 was 701.387367 MB. As was shown in Appendix A, the collinearity between the linkage group and the physical position was relatively disordered at the end of chromosome 6B, and it was not clear whether they were the same QTL.

*Qph.nwafu-4B* (controlling plant height) and *Qtgw.nwafu-4B* (controlling TGW) had an obvious linkage relationship, with a distance of 7–8 cM. The physical location corresponding to this location of *Qph.nwafu-4B* was 30.805339 MB–32.961929 MB, and the physical position corresponding to the location of *Qtgw.nwafu-4B* was 25.847125 MB–26.491497 MB. Guan’s QTL mapping results were marked as *BS00084904_51* and *BS00011338_51* on both sides, and the physical location was 28.954526 MB–66.811785 MB [30]. Cui Fa’s QTL mapping results were marked as *Rht-B1* and *Xmag2055* on both sides, and the physical location was 30.860778 MB–20.741542 MB [70]. Quarrie’s QTL mapping results were marked as *Rht-B1* and *gwm165.1* on both sides, and the physical location was 30.860778 MB–269.948831 MB [42] (The results of previous studies on chromosome 4B and the specific QTL information related to TGW are shown and illustrated in Appendix A). From the QTL mapping results in this study and the above three research results, it was suggested that the confidence interval had this overlap while the confidence interval of this study was the shortest. Based on heredity Doumai/Shi 41875, Li mapped the plant height and TGW. The physical location on chromosome 4B was 46.621203 MB [35], which was not the same QTL. The confidence intervals of *Qph.nwafu-4D*, *Qtgw.nwafu-4D.1*, and *Qgw.nwafu-4D* had clear overlaps and were stably expressed in multiple environments. The mutant at this locus lowered plant height while also decreasing TGW and grain width. *Rht2* had a significant effect on TGW, as previously shown by Mohler et al. [32]. There was a significant overlap in the confidence interval of *Qph.nwafu-5D* controlling plant height and *Qgl.nwafu-5D.3* controlling grain length, with a typical pleiotropism. This locus’s physical position was 466.230408 MB–469.357817 MB, and its additive effect was opposite, so physiological antagonism occurred. The location of *wmc215* targeted by Hai et al. was 472.369175 MB, and that of *gwm212* targeted by Quarrie was 472.630187 MB, which was in line with previous localization results [42,43]. The difference in physical location was 3 MB. Since subgroup D had a large linkage disequilibrium [72], it was impossible to determine whether these loci were the same one. *Qtgw.nwafu-5D.1* controlling TGW and *Qgl.nwafu-5D.1* controlling grain length were located in the region from 38.070293 MB–41.294446 MB, neither of which belonged to the same region of the 5D chromosome, compared with the results of previous studies [35,42,43,73].

### 3.3. Qtgw.Nwafu-4B Molecular Marker Development

Based on the confidence interval of the parental 660K chip marker, two co-segregating KASP markers were developed. Two KASP molecular markers were inserted into the original genetic map, and the genetic map of chromosome 4B maintained a high degree of collinearity. Two KASP molecular markers were inserted into the original genetic map, and the genetic map of chromosome 4B maintained a high degree of collinearity. The primer sequences and typing information of the two molecular markers of KASP_AX-109316968 and KASP_AX-109333198 are shown in Appendix A. *Qph.nwafu-4B* (controlling plant height) and *Qtgw.nwafu-4B* (controlling TGW) had an obvious linkage relationship, with a distance of 7–8 cM. Breeding is based on molecular markers that control plant height and thousand-grain weight to select strains with low plant height and large grain weight.

## 4. Materials and Methods

### 4.1. Plant Materials, Experimental Design, and Investigation of Agronomic Traits

Xinong 1376 is the female parent and Xiaoyan 81 is the male parent. Based on the single-grain transmission method, 198 RIL lines were generated. There were planted in Yangling, Shaanxi province and Nanyang, Henan province, from October 2018 to June 2019 and from October 2019 to June 2020, respectively. A randomized block design (repeated five times, with two rows of districts, 2 m row length, 70 plants per row, and 0.3 m row spacing) was adopted in each experimental site. The other field managements were subject to the same treatment as the local. During the wax maturity period of wheat, five individual plants were sampled in sequence from the fifth plant of each family. Plant height, TGW, grain length and grain width were also measured. By R/lme4 [73], each environment’s agronomic traits were obtained for W-test, and then multiple comparisons of parental traits and calculation of heritability were completed. The heritability of the two traits was calculated by using the formula as follows:*H*^2^ = V_G_∕(V_G_ + V_GY_/y + V_GE_/e + V_E_/nr) × 100%
where y is the number of years, e is the number of environments, and n is the number of repetitions.

The pedigrees of Xinong1376 and Xiaoyan81 are illustrated in Appendix A.

### 4.2. Construction and Evaluation of Genetic Maps

The wheat genomic DNA, with tender wheat leaves as the plant material, was extracted by CTAB, and the quality and quantity of DNA were detected and confirmed. Meanwhile, the DNA of each line was hybridized on the wheat 50K SNP array containing 66,832 markers using Burdock Biotechnology (Beijing, China).

The course of constructing the map was conducted as follows: The BIN function of IciMapping 4.1 [70] was utilized to analyze the markers, and the markers with partial separation rate (*p* < 0.001) and missing rate (>15%) were removed. The Kosambi function with LOD ≥ 5 was applied to group the combined marker groups in JoinMap 4.0; Kosambi mapping of MSTmap [74], according to the clustering results, was used in the markers’ ordination. The flanking sequences of SNPs were BLAST aligned with the genome of IWGSC RefSeq v1.0 (http://www.wheatgenome.org/News/Latest-news/All-IWGSC-data-related-to-the-reference-sequence-of-bread-wheat-IWGSC-RefSeq-v1.0-publicly-available-at-URGI) to obtain their physical locations. The version of BLAST used was 2.2.31 –outfmt 3–num_alignments 5.

### 4.3. Identification of 1BL/1RS Translocation

1RS, applied to identify parents and populations as x-sec-p1/x-sec-p2, respectively, was a specific marker [75]. Xinong1376 was identified as a 1BS/1RS translocation line. 1B/1R genotyping and traits data were stored in Appendix A. Analysis of variance and Duncan’s new multiple range test comparisons based on genotype and trait were conducted.

### 4.4. Detection of Quantitative Loci

IciMapping 4.2 based on the biparental population (BIP) module with the inclusive composite interval mapping (ICIM, http://www.isbreeding.net/software/?type=detail&id=28) was used for QTL mapping on data obtained from different environments. QTL mapping of the phenotypic values in the four environments was carried out. The LOD value was determined in 1000 permutation tests with a = 0.05 (Type I Error) as the parameter, and the background was set and controlled by the positive and negative stepwise regression, with the step width set to 1cM. QTLs were named based on the International Rules of Genetic Nomenclature (http://whea.pw.usda.gov/ggpages/wgc/98/Intro.htm). Mapchart2.3 (https://www.wur.nl/en/Research-Results/Research-Institutes/plant-research/Biometris-1/SoftwareService/Download-MapChart.htm) was used for the drawing of the genetic and QTL mapping. The collinearity drawing of genetic and physical maps, and the calculation of correlation coefficient were conducted by package plotrix (https://cran.r-project.org/src/contrib/Archive/plotrix/) and package (https://github.com/braverock/PerformanceAnalytics) of R software.

### 4.5. Breeding Molecular Marker Development

After obtaining the preliminary QTL mapping results, we anchored the flanking markers to the physical map. In order to develop a competitive allele-specific PCR (KASP) marker that can be used to track stable TGW QTLs, we used the Wheat660K SNP array to further genotype the parents of the Xinong1376/Xiaoyan81 population [63,71]. According to the method of Ma et al. [63], SNPs located in the main QTL interval were selected to develop KASP markers. The developed integrated genetic map of KASP markers was applied to relocate the target QTL.

## 5. Conclusions

In this research, a genetic map covering the entire wheat genome was constructed, with a total of 3136 bin markers, including 15576 SNP markers, and the total length of the linkage map was 4512.79 cM. Except for chromosome 1B, the ends of chromosomes were identified as recombination-rich areas, while the middle areas were recombination-barren. Both the genetic map and the physical map showed a significant correlation when *p* = 0.001. The correlation coefficient ranged from 0.63 to 0.99. There was almost no recombination between 1RS and 1BS. Among 43 QTLs indirectly compared by reference genome, only 13 QTLs were consistent with previous mapping results, and 30 QTLs were defined as new QTLs. Seven environment-stable QTLs were detected in this population, including *Qph.nwafu-4B*, *Qtgw.nwafu-4B*, *Qgw.nwafu-4D*, *Qph.nwafu-4D*, *Qgl.nwafu-5A*, *Qgl.nwafu-5D.2*, and *Qgl.nwafu-6B*. *Qtgw.nwafu-4B*, which influenced TGW, showed an explanation rate of 3.43–6.85% for phenotypic variation, with two co-segregating KASP markers developed, and the physical locations corresponding to KASP_AX-109316968 and KASP_AX-109519968 were 25.888344 MB and 25.847691 MB, respectively, for details, see Figure 2. *Qph.nwafu-4B* (controlling plant height) and *Qtgw.nwafu-4B* (controlling TGW) had an obvious linkage relationship, with a distance of 7–8 cM. The physical location corresponding to this location of *Qph.nwafu-4B* was 30.805339 MB–32.961929 MB, and the physical position corresponding to this location of *Qtgw.nwafu-4B* was 25.847125 MB–26.491497 MB. There is a functional marker (AX-179477460) for the control value of plant height in the *Qph.nwafu-4B* confidence interval, and this locus can be determined to be Rht-B1. The physical locations of *Qph.nwafu-4B*, *Qph.nwafu-4D*, and *Qgw.nwafu-4D* were consistent with previous mapping results. For *Qgl.nwafu-6B*, it couldn’t be accurately determined whether it was a new QTL or not. Two QTL clusters were identified on chromosomes 4B and 4D (Table 3 and Appendix A).

## Figures and Tables

**Figure 1 plants-10-01167-f001:**
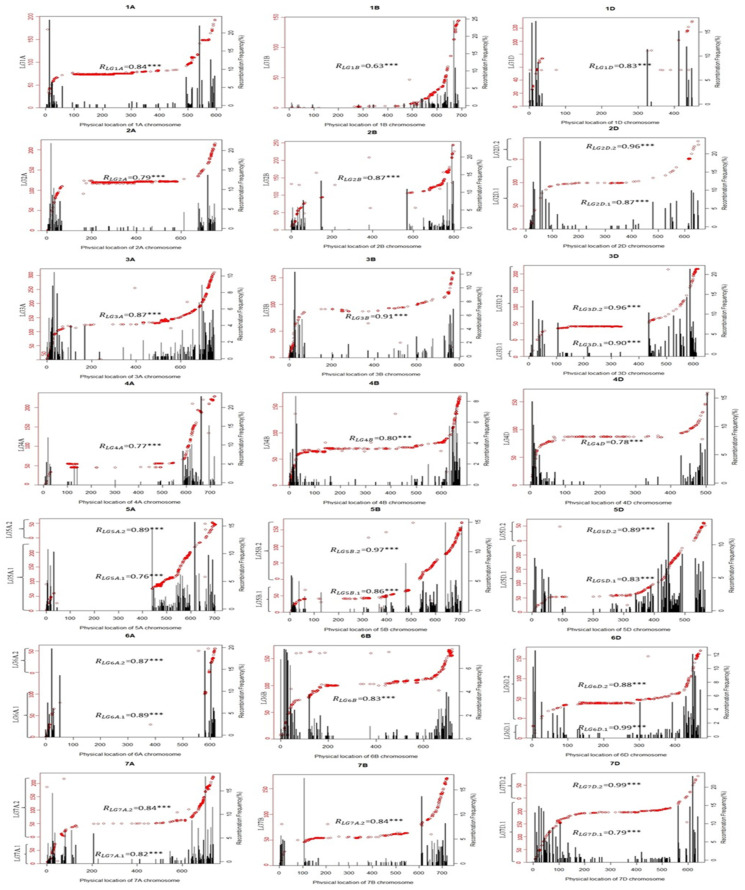
Collinearity analysis of genetic map and reference genome. **NOTE:** The genetic distances of the linkage group are shown as the left Y-axis, the recombination rate of bin markers as the right Y-axis, the physical location of the markers as the x-axis, the collinearity as the red scatter dots, and the recombination rate of bin markers on the reference genome as the black histogram. A, B, D are the three subgroups of common wheat.

**Figure 2 plants-10-01167-f002:**
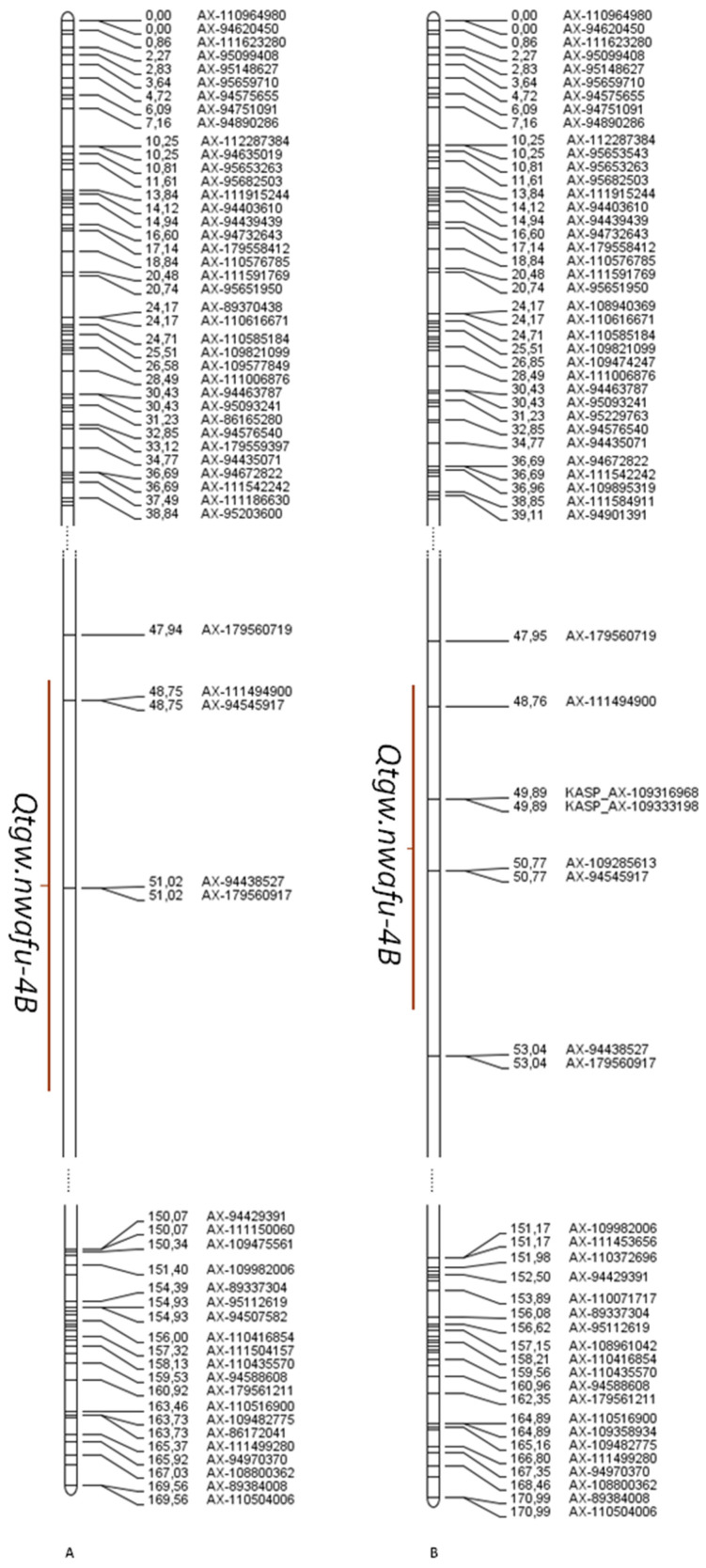
The linkage group corresponding to chromosome 4B. Note: (**A**) represents the original linkage group, and (**B**) the linkage group after the addition of the KASP marker.

**Table 1 plants-10-01167-t001:** Statistical analysis of parent and RIL lines for traits.

Traits	Environment	Xinong1376	Xiaoyan81	Mean ± SD	Minimum	Maximum	*p*-Value	Heritability
Plant height	19NY	65.25	77.75 **	67.08 ± 13.78	32.2	96.8	2.19 × 10^−3^	0.73
	20NY	68.24	81.22 **	80.03 ± 14.43	40.2	109.8	5.66 × 10^−6^	
	19YL	68.36	78.23 **	65.78 ± 12.78	34.6	90.9	6.14 × 10^−4^	
	20YL	72.33	83.25 **	72.24 ± 15.08	38.3	109.2	3.82 × 10^−2^	
Thousand Grain Weight	19NY	41.35 **	36.23	40.72 ± 4.37	27.81	52.19	1.12 × 10^−1^	0.62
	20NY	42.13 **	39.48	42.62 ± 4.51	26.28	51.76	1.24 × 10^−3^	
	19YL	44.32 **	41.75	45.32 ± 4.41	34.22	55.05	2.80 × 10^−2^	
	20YL	46.23 **	42.32	45.21 ± 4.40	29.5	54.83	3.68 × 10^−1^	
Grain length	19NY	7.12 **	6.87	7.23 ± 0.37	6.27	8.04	3.93 × 10^−2^	0.61
	20NY	7.32 **	6.75	7.14 ± 0.35	6.34	8.03	6.79 × 10^−2^	
	19YL	7.51 **	7.24	7.44 ± 0.34	6.68	8.23	1.81 × 10^−2^	
	20YL	7.36 **	7.14	7.51 ± 0.38	6.67	8.51	1.96 × 10^−1^	
Grain width	19NY	3.31	3.21	3.37 ± 0.15	2.88	3.69	1.45 × 10^−3^	0.72
	20NY	3.88 **	3.62	3.45 ± 0.18	2.81	3.83	2.18 × 10^−3^	
	19YL	3.51 **	3.28	3.60 ± 0.16	3.11	3.9	1.38 × 10^−2^	
	20YL	3.66 **	3.42	3.60 ± 0.16	3.16	3.95	3.02 × 10^−3^	

**Note:** ** represents a significant difference between the two parents when *p* = 0.01.

**Table 2 plants-10-01167-t002:** Single-nucleotide polymorphism (SNP) marker statistics about distribution and density on 21 wheat chromosomes derived from crossing between Xinong1376 and Xiaoyan81.

Chromosome	Linkage Group	Length(cM)	Maker Numbers	Bin Number	Insinuation Markers	Maximum Clearance	Average Bin	Bin Density
1A	LG1A	192.66	1064	112	1045	25.68	1.72	0.58
1B	LG1B	144.61	558	118	447	26.86	1.23	0.82
1D	LG1D	130.85	394	42	336	18.01	3.12	0.32
2A	LG2A	215.97	951	140	940	23.46	1.54	0.65
2B	LG2B	244.43	676	173	597	25.44	1.41	0.71
2D	LG2D.1	132.89	161	48	154	25.42	2.77	0.36
	LG2D.2	38.69	77	11	75	10.06	3.52	0.28
3A	LG3A	311.23	1322	285	1301	16.8	1.09	0.92
3B	LG3B	160.61	487	144	458	12.59	1.12	0.9
3D	LG3D.1	17.46	38	8	36	13.71	2.18	0.46
	LG3D.2	214.66	999	87	1026	22.84	2.47	0.41
4A	LG4A	228.42	614	123	592	24.85	1.86	0.54
4B	LG4B	169.56	1185	193	1156	8.57	0.88	1.14
4D	LG4D	166.01	203	77	199	16.52	2.16	0.46
5A	LG5A.1	234.18	969	169	963	16.26	1.39	0.72
	LG5A.2	52.94	139	39	134	9.87	1.36	0.74
5B	LG5B.1	68.44	682	88	675	8.09	0.78	1.29
	LG5B.2	172.4	538	164	529	15.43	1.05	0.95
5D	LG5D.1	223.58	192	119	171	13.69	1.88	0.53
	LG5D.2	61.6	427	42	415	8.03	1.47	0.68
6A	LG6A.1	112.71	154	50	137	20.91	2.25	0.44
	LG6A.2	54.95	161	36	151	17.46	1.53	0.66
6B	LG6B	167.65	852	188	783	7.7	0.89	1.12
6D	LG6D.1	31.08	34	7	34	10.4	4.44	0.23
	LG6D.2	170.88	506	124	497	12.41	1.38	0.73
7A	LG7A.1	75.25	194	76	176	14.91	0.99	1.01
	LG7A.2	225.38	647	201	633	18.9	1.12	0.89
7B	LG7B	170.54	882	129	845	18.19	1.32	0.76
7D	LG7D.1	237.8	453	130	446	15.51	1.83	0.55
	LG7D.2	85.38	17	13	16	24.96	6.57	0.15
1st homologous	3	468.12	2016	272	1828	26.86	1.72	0.58
2nd homologous	4	631.98	1865	372	1766	25.44	1.7	0.59
3rd homologous	4	703.96	2846	524	2821	22.84	1.34	0.74
4th homologous	3	563.99	2002	393	1947	24.85	1.44	0.7
5th homologous	6	813.14	2947	621	2887	15.43	1.31	0.76
6th homologous	4	537.27	1707	405	1602	20.91	1.33	0.75
7th homologous	5	794.35	2193	549	2116	24.96	1.45	0.69
A genome	10	1703.69	6215	1231	6072	25.68	1.38	0.72
B genome	8	1298.23	5860	1197	5490	26.86	1.08	0.92
D genome	12	1510.87	3501	708	3405	25.44	2.13	0.47
TOTAL	30	4512.79	15576	3136	14967	26.44	1.44	0.69

**Table 3 plants-10-01167-t003:** Full genomic QTL mapping results of plant height and grain related traits in the F8 RIL lines between Xinong1376 and Xiaoyan81.

Trait	QTLs Name	Environment	Position	LOD	PVE (%)	Add	Left and Right Marker	Interval	Physical Interval	Reference
PH	*Qph.nwafu-2D.1*	19YL	17	10.7	3.73	−3.77	AX-111561744/AX-179557748	16.5–20.5	23.416254/28.417456	(Zhai et al., 2016)
PH		20YL	17	8.82	3.3	−3.75	AX-111561744/AX-179557748	16.5–20.5	23.416254/28.417456	
PH	*Qph.nwafu-2D.2*	19YL	103	53.34	33.08	11.29	AX-94570302/AX-109998182	102.5–103.5	413.778968/425.474614	
PH	*Qph.nwafu-4B*	19NY	59	7.39	10.23	4.93	AX-179477460/AX-110984065	58.5–59.5	30.805339/32.961929	(Mohler et al., 2016)
PH		20NY	59	9.36	13.76	5.9	AX-179477460/AX-110984065	58.5–59.5	30.805339/32.961929	
PH		19YL	59	23.33	9.32	6.28	AX-179477460/AX-110984065	58.5–59.5	30.805339/32.961929	
PH		20YL	59	26.85	12.26	7.62	AX-179477460/AX-110984065	58.5–59.5	30.805339/32.961929	
PH	*Qph.nwafu-4D*	19NY	62	17.27	27.08	−7.69	AX-86170701/AX-89445201	61.5–62.5	18.781207/19.459614	(Zhang et al., 2013)
PH		20NY	62	16.78	27.09	−7.93	AX-86170701/AX-89445201	61.5–62.5	18.781207/19.459614	
PH		19YL	62	40.17	20.11	−8.84	AX-86170701/AX-89445201	61.5–62.5	18.781207/19.459614	
PH		20YL	62	42.21	23.71	−10.16	AX-86170701/AX-89445201	61.5–62.5	18.781207/19.459614	
PH	*Qph.nwafu-5B*	20YL	55	4.32	1.49	2.52	AX-109908739/AX-86174612	54.5–55.5	422.122099/425.671678	
PH	*Qph.nwafu-5D*	20YL	190	46.92	29.23	−11.18	AX-94390434/AX-110033637	189.5–190.5	466.230408/469.357817	(Quarrie et al., 2005; Hai et al., 2008)
PH	*Qph.nwafu-6B.1*	20YL	139	6.19	2.19	3.07	AX-109987590/AX-86162252	137.5–139.5	687.177084/688.20385	
PH	*Qph.nwafu-6B.2*	19YL	160	4.15	1.36	2.27	AX-110632551/AX-109509377	159.5–160.5	712.125253/711.370298	
TGW	*Qtgw.nwafu-2A*	20YL	186	2.6	5.08	0.95	AX-95103231/AX-94508212	185.5–186.5	733.854404/734.347961	(Cui et al., 2014)
TGW	*Qtgw.nwafu-2B*	20YL	101	4.24	10.14	1.34	AX-108905289/AX-95235626	95.5–106.5	153.585606/568.468886	(Li et al., 2018)
TGW	*Qtgw.nwafu-3A*	19YL	133	2.7	4.14	−0.89	AX-179477407/AX-94457296	132.5–134.5	457.796943/431.074614	
TGW	*Qtgw.nwafu-4B*	20NY	51	4.18	3.43	1.23	AX-111494900/AX-94438527	48.5–53.5	25.847125/26.491497	[67]
TGW		19YL	51	4.37	6.85	1.18	AX-111494900/AX-94438527	49.5–53.5	25.847125/26.491497	
TGW		19NY	52	2.85	5.02	1.06	AX-94438527/AX-110383634	48.5–55.5	26.491497/28.71668	
TGW	*Qtgw.nwafu-4D.1*	19NY	60	5.87	9.73	−1.44	AX-89703298/AX-86170701	56.5–60.5	16.926631/18.781207	(Mohler et al., 2016)
TGW		20NY	60	6.2	5.25	−1.48	AX-89703298/AX-86170701	56.5–60.5	16.926631/18.781207	
TGW	*Qtgw.nwafu-4D.2*	19YL	111	3.54	5.55	−1.03	AX-111926032/AX-94818797	107.5–112.5	476.884228/477.371597	
TGW	*Qtgw.nwafu-5A*	19YL	44	7.18	11.94	1.51	AX-95510385/AX-95117188	43.5–45.5	698.508129/702.466804	
TGW		20YL	44	3.51	6.99	1.11	AX-95510385/AX-95117188	43.5–45.5	698.508129/702.466804	
TGW	*Qtgw.nwafu-5D.1*	19NY	37	5.51	9.32	1.4	AX-111543112/AX-110576074	34.5–38.5	38.070293/41.294446	
TGW		20NY	37	14.79	14.24	2.43	AX-111543112/AX-110576074	35.5–38.5	38.070293/41.294446	
TGW	*Qtgw.nwafu-5D.2*	20NY	46	6.46	5.56	−1.51	AX-111019963/AX-110085499	44.5–49.5	42.928674/44.192407	
TGW	*Qtgw.nwafu-5D.3*	19YL	81	3.66	5.83	1.05	AX-110867187/AX-108827297	79.5–81.5	369.202139/370.064947	
TGW	*Qtgw.nwafu-6A*	20YL	29	3.06	6.26	−1.06	AX-109431286/AX-109358667	27.5–30.5	606.979733/608.046298	(Cui et al., 2014)
GL	*Qgl.nwafu-1A*	19YL	150	3.3	3.11	−0.06	AX-95682344/AX-108726119	148.5–150.5	572.350803/572.658176	(Mir et al., 2012)
GL	*Qgl.nwafu-1B.1*	20YL	0	3.62	4.51	0.09	AX-94835306/AX-179476279	0–0.5	59.471177/94.978091	
GL	*Qgl.nwafu-1B.2*	19NY	65	3.51	4.49	0.08	AX-94650293/AX-112288501	64.5–66.5	640.845515/641.632325	
GL		19YL	65	5.23	5.14	0.08	AX-94650293/AX-112288501	64.5–65.5	640.845515/641.632325	
GL	*Qgl.nwafu-3A*	20YL	135	5.71	7.61	−0.1	AX-94426283/AX-110122062	134.5–136.5	511.755031/510.853056	
GL		20NY	137	3.78	6.88	−0.09	AX-179557644/AX-94387510	136.5–137.5	541.482465/540.048345	
GL	*Qgl.nwafu-4A*	20NY	49	4.41	9.11	0.1	AX-111251110/AX-179476673	46.5–53.5	407.389107/129.089816	
GL		19NY	50	4.57	6.07	0.1	AX-111251110/AX-179476673	47.5–52.5	407.389107/129.089816	
GL	*Qgl.nwafu-4B.1*	19YL	51	4.28	4.17	0.07	AX-179476673/AX-110173140	47.5–52.5	129.089816/140.310606	
GL	*Qgl.nwafu-4B.2*	19YL	68	5	4.94	−0.08	AX-109507847/AX-109427900	67.5–69.5	114.952789/161.548436	(Wang et al., 2010)
GL		20YL	68	4.03	5.11	−0.08	AX-109507847/AX-109427900	67.5–69.5	114.952789/161.548436	
GL	*Qgl.nwafu-4D*	19YL	16	2.67	2.55	0.05	AX-108892806/AX-109447997	15.5–18.5	6.598631/7.048661	
GL	*Qgl.nwafu-5A*	19NY	44	6.93	9.59	0.12	AX-95510385/AX-95117188	43.5–44.5	698.508129/698.508129	
GL		19YL	44	6.47	6.72	0.09	AX-95510385/AX-95117188	43.5–44.5	698.508129/700.34701	
GL		20YL	44	5.8	7.73	0.1	AX-95510385/AX-95117188	43.5–45.5	698.508129/700.34701	
GL	*Qgl.nwafu-5B*	19YL	6	2.75	2.56	−0.05	AX-112288130/AX-95631525	5.5–6.5	6.654131/8.917454	
GL	*Qgl.nwafu-5D.1*	20NY	37	3.13	5.76	0.08	AX-111543112/AX-110576074	33.5–39.5	38.070293/41.294446	
GL	*Qgl.nwafu-5D.2*	19YL	82	6.13	5.99	0.08	AX-111496494/AX-109707913	81.5–84.5	370.135626/379.028214	
GL		20YL	82	6	7.75	0.1	AX-111496494/AX-109707913	81.5–84.5	370.135626/379.028214	
GL		19NY	89	4.36	5.62	0.09	AX-110558491/AX-111903917	88.5–91.5	385.893875/386.126855	
GL	*Qgl.nwafu-5D.3*	20YL	191	7.45	9.83	0.11	AX-110033637/AX-110830424	190.5–191.5	469.357817/469.523881	
GL	*Qgl.nwafu-5D.4*	19YL	218	6.85	7.84	0.09	AX-110777538/AX-111512534	215.5–221.5	485.909071/491.01105	
GL	*Qgl.nwafu-6B*	19YL	162	8.79	8.95	0.1	AX-110287286/AX-111572797	161.5–162.5	712.125253/712.245125	(Li et al., 2018)
GL		19NY	167	7.88	10.73	0.13	AX-89379712/AX-94499484	166.5–167	704.884934/718.376276	
GL		20NY	167	3.55	6.68	0.09	AX-89379712/AX-94499484	166.5–167	704.884934/718.376276	
GL		20YL	167	7.73	10.33	0.12	AX-89379712/AX-94499484	166.5–167	704.884934/718.376276	
GW	*Qgw.nwafu-2B*	20YL	94	3.86	6.13	0.04	AX-109423066/AX-108990832	93.5–94.5	152.611396/153.128588	
GW	*Qgw.nwafu-2D*	19NY	12	2.55	4.2	0.04	AX-179477408/AX-111367738	11.5–12.5	20.768547/21.405473	(Huang et al., 2006; Guan et al., 2018; Wu et al., 2015)
GW	*Qgw.nwafu-3A*	19YL	311	3.18	4.66	0.04	AX-110915909/AX-110475339	308.5–311	746.360221/749.849798	(Lee et al., 2014)
GW	*Qgw.nwafu-4B.1*	19NY	51	4.15	6.95	0.05	AX-111494900/AX-94438527	49.5–54.5	25.847125/26.491497	
GW		20NY	51	3.5	6.85	0.05	AX-111494900/AX-94438527	48.5–54.5	25.847125/26.491497	
GW	*Qgw.nwafu-4B.2*	20YL	68	3.91	6.23	0.04	AX-109507847/AX-109427900	67.5–68.5	114.952789/161.548436	(Wang et al., 2010)
GW	*Qgw.nwafu-4B.3*	19YL	77	9.23	15.24	0.07	AX-179559104/AX-95658798	76.5–77.5	520.214474/523.447693	
GW	*Qgw.nwafu-4D*	20NY	60	3.2	6.32	−0.05	AX-89703298/AX-86170701	59.5–61.5	16.926631/18.781207	
GW		19NY	61	4.15	7.22	−0.05	AX-86170701/AX-110572006	59.5–61.5	18.781207/19.179341	
GW		19YL	63	7.93	12.12	−0.06	AX-86170701/AX-89445201	61.5–64.5	18.781207/19.459614	
GW		20YL	63	6.37	10.37	−0.05	AX-86170701/AX-89445201	61.5–63.5	18.781207/19.459614	
GW	*Qgw.nwafu-5D*	19NY	163	3.5	6.01	0.04	AX-109317498/AX-109855976	159.5–166.5	448.686533/449.292436	
GW	*Qgw.nwafu-6D*	20NY	4	2.51	5.8	0.05	AX-111594857/AX-109406081	0–12.5	12.650045/8.255713	

Note: PH, TGW, GL, and GW represent plant height, thousand-grain weight, grain length, and grain width, respectively. Reference represents that the confidence interval of this study overlaps with that of previous studies.

## Data Availability

Please refer to suggested Data Availability Statements in section “MDPI Research Data Policies” at https://www.mdpi.com/ethics.

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
