# Peer review of "Utilization of a Wheat50K SNP Microarray-Derived High-Density Genetic Map for QTL Mapping of Plant Height and Grain Traits in Wheat"

_plants, 2021, doi:10.3390/plants10061167_

Round 1

Reviewer 1 Report

The manuscript by Dongyun Lv et al. has shown on the utilization of a Wheat50K SNP microarray-derived high-density genetic map for QTL mapping of plant height and grain traits in wheat. However, this article need to revise major revision as follows:

Page 1, Abstract: Authors need to reduce some unimportant parts and to describe on the prospects of the results obtained in the study as an ending remarks.

Page 2~3, Introduction: There are too many of 61 lines in ‘Introduction’ including some parts of redundancy and missing. Authors need to trim some parts and to include about KASP marker.

Page 3, Line 93~140: ‘Materials and Methods’ should be moved after Discussion.

Page 3, Line 94~106: ‘2.1. Test materials and phenotype treatment’ --- Authors need to change as follows: 2.1. Plant materials, experimental design, and investigation of agronomic traits’

Page 3, Line 95: Two fine varieties, Xiaoyan81 and Xinong1376, and their two derivative lines F2:8 containing 198 RILs families --- Authors should describe the history of the F8 RIL population in detail.

Page 4, Line 108: test materials --- plant materials

Page 5, Line 142: Phenotypic data analysis --- Agronomic traits analysis.  * phenotypic data or phenotype should be changed to agronomic traits or traits through manuscript.

Page 5, Line 156: Table 1 Statistic analysis of parent and RIL family for phenotype --- Table 1 Statistic analysis of parent and RIL family for traits

Page 6, Line 166: The total lemgth --- The total length

Page 10, Line 220: in Xinong1376/ Xiaoyan81 derived population --- in the F8 RIL population between Xinong1376 and Xiaoyan81

Page 14, Line 280: 198 RIL populations --- 198 F8 RIL populations

Page 15, Line 308: 4.2. Mapping results and previous --- Rename this subtitle well.

Page 18, Line 380: Authors need to describe on the prospects of the results obtained in this study as an ending remarks.

Author Response

The changes have been made as required, and the changes have been marked in yellow.

Reviewer 2 Report

p1/Abstract, "when p=0.001", I suppose p<0.001
probably just leave out p=0.001, significant is enough
"correlation coefficient between 0.63 and 0.99" , I suppose
when comparing the various chromosomes
p2/58 "largest crops genome" => "largest crop genome"
p2/59 "repetitive sequence TE" => repetitive sequences such as transposable elements
p5/143 "extremely significant" => would not use "extremely"

figure1 could get a better figure resolution
p4/115: "were BLAST aligned" => sloppy wording, missing also parameters and BLAST version
p4/127: "with a=0.05 as the parameter" => please explain what this
parameter is
which other studies and results have been obtained with the two
wheat lines in other studies and were there limitatinos and shortcomings in those
to motivate the current study

Author Response

The changes have been made as required, and the changes have been marked in red.

Round 2

Reviewer 2 Report

remarks were properly addressed